# A Modified PINN Approach for Identifiable Compartmental Models in Epidemiology with Application to COVID-19

**DOI:** 10.3390/v14112464

**Published:** 2022-11-07

**Authors:** Haoran Hu, Connor M. Kennedy, Panayotis G. Kevrekidis, Hong-Kun Zhang

**Affiliations:** Department of Mathematics and Statistics, University of Massachusetts Amherst, Amherst, MA 01003, USA

**Keywords:** network dynamics, COVID-19, PINNs, wavelets

## Abstract

Many approaches using compartmental models have been used to study the COVID-19 pandemic, with machine learning methods applied to these models having particularly notable success. We consider the Susceptible–Infected–Confirmed–Recovered–Deceased (SICRD) compartmental model, with the goal of estimating the unknown infected compartment *I*, and several unknown parameters. We apply a variation of a “Physics Informed Neural Network” (PINN), which uses knowledge of the system to aid learning. First, we ensure estimation is possible by verifying the model’s identifiability. Then, we propose a wavelet transform to process data for the network training. Finally, our central result is a novel modification of the PINN’s loss function to reduce the number of simultaneously considered unknowns. We find that our modified network is capable of stable, efficient, and accurate estimation, while the unmodified network consistently yields incorrect values. The modified network is also shown to be efficient enough to be applied to a model with time-varying parameters. We present an application of our model results for ranking states by their estimated relative testing efficiency. Our findings suggest the effectiveness of our modified PINN network, especially in the case of multiple unknown variables.

## 1. Introduction

On 31 December 2019, 27 cases of pneumonia were reported in Wuhan City, Hubei Province in China. The cause was identified on 7 January 2020, and was subsequently termed SARS-CoV-2 (the virus) and COVID-19 (the disease) by the World Health Organization (WHO) [1]. The disease subsequently grew to the point that the WHO officially declared it a pandemic on 11 March 2020 [2]. As of 31 May 2022, a cumulative total of 526,558,033 cases and 6,287,117 deaths attributed to the disease had occurred [3]. The sheer scale of the disease’s development prompted research to combat it across the globe. A proper modeling of the disease has been crucial not only in evaluating the behavior of the virus, but also towards policy decisions made in response to it [4,5]. An extensive evaluation of social-distancing measures and other nonpharmaceutical interventions found that these approaches were effective. These results supported the adoption of these policies on numerous occasions throughout the pandemic [6,7,8]. More recently, the role of vaccination has also been considered in the relevant models [9].

One of the greatest issues in studying the disease has been the difficulty in estimating its spread. Even simply getting an accurate estimate of the current number of infected individuals has been extremely difficult [10]. Accurate counts of the infected population are crucial. They serve as a guiding tool on policy decisions from the distribution of testing resources, to the allocation of treatment materials, to the severity of lockdown procedures. Limited testing supplies in the early stages and the frequency of asymptomatic cases have caused major information gathering difficulties [11]. The accurate estimation of certain parameters used in modeling the disease, notably the base reproduction rate and case confirmation rate, are also useful in guiding policy. The parameters’ true values, however, are dependent on these unknown infected cases. There are some methods, however, to infer these unknown quantities from limited available data; see, e.g., [12] for a recent discussion. We focus here specifically on the usage of compartmental epidemiological models in conjunction with the usage of the widely applicable and highly successful technology of the so-called physics-informed neural networks (PINNs) (see [13] for a recent review thereof) to estimate these quantities.

The classic SIR ODE model is, arguably, the most well-known compartmental model. It separates the population into susceptible, infected, and removed (recovered) groups, with the model’s origins tracing back to the seminal work of [14]. In the almost century that has followed there has been a wide variety of variations of the model to fit the particular features of different diseases [15]. In the case of COVID-19, several different modifications have been considered to more accurately describe its development. The SEAHIR model (involving susceptible, exposed, asymptomatic, hospitalized, infected, and recovered populations) incorporates the effect of social isolation measures [16]. Meanwhile, the SIRSi (susceptible, infected, recovered, sick) approach models temporary immunity that wanes over time [17]. There have been numerous other proposals, involving different numbers of components (and other features such as, e.g., age stratification [18] and spatial distribution [19]). More complex models do run the risk of their unknown quantities being difficult to estimate, or in some cases even impossible [20]. If we want to estimate these unknowns, we need a model that reflects the practical realities of COVID-19, the effects of isolation [6], and the prevalence of asymptomatic cases [10,11], while still allowing estimations.

We adopt the usage of the SICRD model implemented in [21]. The susceptible, infected, and recovered groups are similar to the standard SIR setting discussed above. We now also consider two additional compartments, the confirmed and death cases, which we assume to be the only directly known variables. We adopt this model as it incorporates the noteworthy effects of testing and quarantining. It also crucially splits the infected population into unknown/known compartments, as well as includes data on fatalities. The latter is an extremely important piece of accessible information as we note in Section 2.3.2. Indeed without it, it would actually be impossible to estimate the number of infected individuals, or most of the important parameters from confirmed case counts alone. The relevance of the inclusion of the death data (as well as potentially and more recently, the data on hospitalizations) is a feature that some of the present authors have argued in various earlier works [12,18]. The SEAHIR model has similar advantages and much of the analysis below could be adapted to that model as well (an avenue that we do not pursue herein).

We propose the usage of a sophisticated neural network approach to estimate the unknown cases and parameters, i.e., a methodology that has proved extremely successful in a variety of applications within scientific computing [13,22]. A key component in the usage of a neural network over traditional regression techniques is in their improved flexibility. Regression methods such as those in [23] may only model the data as exactly following a specified model. A neural network has many more adjustable parameters, allowing for a more flexible modeling of a given problem. Indeed, in this vein, there have been some quite promising recent results in the usage of neural networks in disease modeling. The work of [24] effectively forecasts COVID-19 spread using a combination of an ensemble neural network with a fuzzy logic system. The usage of a graph neural network leveraged human mobility information to improve forecasting of COVID-19 in Germany [25]. Meanwhile, the spread of influenza in the U.S. has been modeled using a recurrent neural network to better reflect the time-sequential nature of the viral spread [26]. The usage of a neural network is also quite natural for us, as our model is particularly well suited for an adaptation of the “Physics Informed Neural Network” (PINN) approach [27].

The principle of PINNs was developed for the estimation of unknown quantities in a system adhering to a physical law, generally a nonlinear PDE (or lattice dynamical equation [28]) that the system is assumed to obey [27]. The effectiveness of this approach has been substantial across a wide variety of disciplines, notably having excellent compatibility with other techniques such as the previously mentioned graph and recurrent neural networks [22]. We note there have been several other attempts to apply the PINN concept to studying COVID-19. The work of [29] considered the SIR model. They suitably modified the dynamical equations thereof through a nonlinear transformation, aiming to leverage neural networks to identify the (assumed-to-be) linear dependence of the infection rate and subsequently performed short-term predictions. This has some interesting parallels to our proposals and analysis below, although we seek to avoid some of the relevant assumptions on the reliability of the total case counts or of the linear time-dependence of the infection rate. Meanwhile, the work of [30] also attempted a similar inference of parameters to our work and considered the more nuanced susceptible–exposed–infected–recovered–susceptible model. This attempt, however, also considered all variables to be known. Yet another effort in this vein of leveraging data-driven techniques in epidemiological problems can be found in the interesting work of [31]. In this paper, the authors sought to discover the associated dynamic models directly from data in the case of childhood diseases such as chickenpox, rubella, and measles, with partial success.

The novelty of our work comes from the substantial difficulty of having unknown variables in our system and the modifications of the PINN technique we make to overcome this. The standard PINN directly uses the governing equations to define its loss term, with no further reformulation specific to a given model. We found that in our case, with several unknown variables, the standard PINN consistently converged to incorrect values for all parameters and variables. We address this by introducing a new approach to the training loss. We derive a new set of terms for our loss function in Section 2.5. The new loss terms are structured to reduce the number of unknowns that must be simultaneously considered in a single term. This avoids errors due to too many degrees of freedom in the network’s attempted estimation. The result is a network with rapid and accurate convergence. We achieve very promising results, even with a simple feed-forward network architecture. This suggests the relevance of further research into other architectures, such as recurrent networks to improve performance, or an application to a graph neural network to incorporate human mobility [19,32]. The exact form of our new loss is specific to the SICRD model, but the principle of deriving new equations for the purpose of improved stability of the training is far broader. The concept could be applied to the training loss of any machine learning system, so long as there exist some governing equations for the considered data. Thus, any such analysis could see improvements in the network’s convergence, stability, and efficiency with appropriately derived loss terms from the governing equations. In summary, we propose the usage of a neural network approach to avoid the more substantial restraints of traditional regression methodologies. We use the PINNs among the former to connect to a widely used tool. Finally, we provide an approach that leads to an improvement of the training loss of the latter and whose principle can be generalized to other systems within this class (of relevance to epidemiological applications).

We begin in Section 2.1 and Section 2.2 by explicitly defining the SICRD model, as well as explaining the general concepts of the model and the sources of our data. In Section 2.3, we consider the question of whether the unknown variables and parameters may be estimated from the partially known information. We formally address this by showing our model is identifiable, meaning the unknown parameters and unknown variables can be uniquely determined by the known variables. We thus verify we are in a regime where the estimation goal is possible. It is also shown that several variations of our model are not identifiable, which justifies our particular choice over several considered alternatives. In Section 2.4, we address the issue of potential noise in our collected data with the use of a wavelet transform, which separates the signal into key features and the noisy component caused by smaller scale fluctuations. This filtering also smooths the data, which improves our ability to apply an ODE model to it. With the model defined and the data appropriately processed, Section 2.5 explicitly presents our novel loss function for the network along with a corresponding explanation for its usage.

We then demonstrate the neural network’s effectiveness by running estimations on the ideal case, with simulated data that explicitly obey our model in Section 3.1. We briefly review the results of denoising via the wavelet transform in Section 3.2. Then, in Section 3.3 we demonstrate the network’s ability to estimate the unknown infected population and several unknown parameters given real data reported in U.S. states. In Section 3.3.1, we also demonstrate that, due to the high efficiency in the network’s calculations, it may also be used to perform estimates using a model with time-varying parameters as opposed to simply assuming constant parameters, and no particular restrictions on the form of time-dependence are necessary, extending the work of other attempts to allow time-varying parameters; see, e.g., [21] or [29]. Our results suggest that when the parameters are assumed to vary, the variance is substantial enough that over any longer time frame, constant parameter assumptions would be unrealistic.

In Section 3.3.2, we propose a ranking of states, according to how well the states are conducting testing by calculating the ratio of the (estimated) infected population over the confirmed population. The intention is to give a more precise estimate on how effectively testing is being conducted in a region, rather than simply looking at the per capita number of infections. This can highlight population centers where infections are low, as the disease has only entered it recently, but testing is low even relative to the number of infections. If a region with poor testing can be recognized early, then the problem can be addressed before it reaches a major outbreak. Finally, in Section 4, we briefly review our results and propose several possible directions of future research. We discuss some of the limitations of the model and network, as well as several methods to expand the work to address those limitations.

## 2. Materials and Methods

### 2.1. The SICRD Model

In modeling the development of COVID-19, we focused on U.S. states as our population centers due to the relative ease of access to epidemiological statistics. In each state, we used a modification of the SIR model [21] by introducing two new population compartments: the death cases and the confirmed cases [21]. This gives the SICRD model, a compartmental model in which the population is divided into susceptible (S), infectious (I), confirmed (C), recovered (R), and dead (D) individuals; see also Figure 1. Note that each compartment refers to the current count and not the total cumulative number of cases.

This model reflects the real-life situation in which an infectious person may recover without receiving a formal diagnosis, as well as accounting for the effect of testing for the disease and quarantining. The model is defined explicitly by the following ODE system:(1)$$\begin{array}{ccc} \dot{S} & = & -\frac{R_{0}}{T_{L}}\frac{SI}{N}\\ \dot{I} & = & \frac{R_{0}}{T_{L}}\frac{SI}{N}-\frac{1}{T_{L}}I\\ \dot{C} & = & \frac{\alpha }{T_{L}}I-\frac{1}{T_{R}}C\\ \dot{D} & = & \frac{\beta (1-\alpha )}{T_{L}}I+\frac{\beta }{T_{R}}C\\ \dot{R} & = & \frac{\left(1-\beta )(1-\alpha \right)}{T_{L}}I+\frac{1-\beta }{T_{R}}C \end{array}$$

Here, N=S+I+C+D+R is the total population size. $R_{0}$ is the basic reproductive number, which refers to the the average number of cases directly infected by one infectious case in a completely susceptible population. $T_{L}$ is the average number of days from first becoming infectious to confirmation, recovery, or death. $T_{R}$ is the average number of days from being confirmed as infected to recovery or death. α is the proportion of confirmed cases among all cases transferred from the unconfirmed infectious state. Finally, β is the fatality rate. The deaths (*D*) and recoveries (*R*) stem from either the infectious compartment (*I*) or from the confirmed infection one (*C*). We also note that we assume individuals who have been confirmed to be infectious sufficiently quarantine and do not infect any further individuals.

Our goal is to estimate the unknown infected population I(t), along with the unknown parameters α, β, and $R_{0}$. We wish to perform an estimation using only our accessible data, the count of confirmed cases C(t), deaths D(t), and the more directly estimable time scale parameters $T_{L}$ and $T_{R}$. The reasons for this precise formulation of the SICRD model are discussed further in Section 2.3.

We note that other, much more complex compartmental epidemiological models with more compartments such as those presented in [18] were also considered; here, inspired by the latter work, we present a reduced version of the model for reasons also relating to identifiability, as discussed below. Furthermore, this more straightforward model was chosen to test the capabilities of our neural network approach without added complications, while the testing of more detailed models is left to be considered in future work.

#### SICRD with Time-Dependent Parameters

The assumption that the parameters in Equation (Equation 1) are constant (i.e., time-independent) is a highly restrictive and unrealistic one for long-enough time scales. In a real-world scenario, many of these parameters are changing as time passes, due to factors such as mutations of the virus and a shifting government policy; the latter is well-known to modify social interactions in the case of nonpharmacological interventions and hence affect factors such as $R_{0}$ [6,18]. In this case, we update the model in (Equation 1) to replace $\alpha ,R_{0},$ and β with $\alpha \left(t\right),R_{0}\left(t\right),$ and β(t), while $T_{L}$ and $T_{R}$ are left fixed as they are relatively well-known, stable parameters. In the case of mutations, it is not unreasonable to expect variations of these parameters too, but for the cases under consideration, we expect such variations to be small and anyway secondary to the above effects. This gives the alternate model:(2)$$\begin{array}{ccc} \dot{S} & = & -\frac{R_{0}\left(t\right)}{T_{L}}\frac{SI}{N}\\ \dot{I} & = & \frac{R_{0}\left(t\right)}{T_{L}}\frac{SI}{N}-\frac{1}{T_{L}}I\\ \dot{C} & = & \frac{\alpha (t)}{T_{L}}I-\frac{1}{T_{R}}C\\ \dot{D} & = & \frac{\beta (t)(1-\alpha (t))}{T_{L}}I+\frac{\beta (t)}{T_{R}}C\\ \dot{R} & = & \frac{\left(1-\beta (t))(1-\alpha (t)\right)}{T_{L}}I+\frac{1-\beta ((t)}{T_{R}}C \end{array}$$

To perform an estimation on the vector of parameters $p\left(t\right)=(R_{0}\left(t\right),\alpha \left(t\right),\beta \left(t\right))$, we first estimate a constant value for the parameters over a time interval of length $\Delta_{t}$. We then perform the estimation of $p(\Delta_{t}+t)$ with a simple “rolling window” approach, where we take the constant parameter estimation over the interval $\left[t,t+\Delta_{t}\right]$ to approximate $p(\Delta_{t}+t)$. This method allows an estimation of the parameter values which does not presume any particular form for their time dependence. This expands on the interesting, very recent work of [29]. The primary potential issue is the costliness of performing so many estimations, but we demonstrate in Section 3.3 that our network performs efficiently enough to make this scheme feasible.

### 2.2. Data Set

All nonsimulated data on COVID-19’s development used in this paper were pulled from the tool developed in [33]. The tool aggregates data from a variety of data sources including the WHO, each state’s individual department of health, and the CDC. The data used included reports on case counts, active cases, and deaths within each individual U.S. state. The usage of these data is expanded upon in Section 3.2, Section 3.3.1 and Section 3.3.2.

### 2.3. Identifiability

#### 2.3.1. Identifiability Definitions

In order to reasonably use Equation (Equation 1), we need to verify the identifiability of the model. Identifiability is the ability to uniquely identify the model parameters from the known variables. We know from the results of [18,20,34] that many compartmental epidemiological models have potential issues with identifiability. To be concrete, we recall the precise definition of the term.

Consider an *n*-dimensional ODE system. We let $p\in \Omega_{p}\subset R^{n_{p}}$ be the vector of constant parameters for the ODE system where $\Omega_{p}$ is our allowable parameter space. We note that the initial values of the variables in the system are also considered parameters. We let $m\left(t,p\right)\in R^{n_{m}}$ be all the assumed known (measured) variables of the system and $h\left(t,p\right)\in R^{n_{h}}$ be all the assumed unknown (hidden) variables of the ODE system, with $n_{m}+n_{h}=n$. Then, the ODE system can be represented by [20]:(3)$$\begin{array}{c} \dot{m}=f\left(m,h,p\right)\\ \dot{h}=g\left(m,h,p\right). \end{array}$$

Note that the same ODE system may be treated with different separations into *m* and *h* depending on what variables are assumed known. Given such a system and a choice of known variables, we say that it is structurally globally identifiable if
(4)$$m\left(t,p\right)=m\left(t,\hat{p}\right),\forall t\geq 0\Rightarrow \hat{p}=p.$$

The interpretation is that the measured variables uniquely determine the values of the constant parameters for the ODE system. In such a case it is not possible for two distinct choices of parameters to give precisely the same values for the measured variables. Note though, from a practical perspective, that this does not disallow the possibility of *p* and $\hat{p}$ with very different values, resulting in $m\left(t,p\right)\approx m\left(t,\hat{p}\right)$. Whether a narrow range for the parameter values can be determined from an uncertain value for *m* is the question of whether the system is *practically identifiable* or not. Basic global identifiability needs to be verified first though, as the practical identifiability of a system is dependent on the particular values chosen for the parameters and the range of the variables [20]. For an interesting example where the range of variables may play a crucial role in the practical identifiability, see, e.g., the very recent work of [35].

It is also possible for a system to fail to be globally identifiable, but still be “locally identifiable” meaning that no unique choice of parameters may be determined from m(t,p), but that there are finitely many choices of *p* that can be made. We obtain in Section 2.3.2 that no cases of local but not global identifiability for any parameters are found for our considered systems.

For Equation (Equation 1), we always assume that *C* and *D* are known and thus m(t,p)=(C,D). *I* is, by definition, not directly known, while *S* and *R* both require knowledge of *I* to be known and thus h(t,p)=(I,S,R). (Reported data on recoveries are available but this information is only on recoveries from confirmed cases and thus is not actually the same quantity as *R* in our model.) We want to guarantee that our model at least satisfies structural global identifiability, meaning it is possible to estimate the parameters from *C* and *D* alone.

#### 2.3.2. Numerical Results on Identifiability

The precise calculation of structural identifiability results for a given ODE system is usually infeasible to perform by hand for all but the simplest of systems. There exist many software applications to perform this type of calculation, such as the differential algebra techniques of [34]. We elected to use the SIAN (Structural Identifiability ANalyser) package to test the identifiability of our model and several slight variations of the model [36]. The code runs a Monte Carlo algorithm to verify both local and global identifiability for an ODE system to within a high degree of certainty (>99%). The following models were tested using the code.

The first model is the system of equations given in (Equation 1) without modifications. The second model matches (Equation 1) for $\dot{S},\dot{I},$ and $\dot{C}$ with the following equations for $\dot{D}$ and $\dot{R}$
(5)$$\begin{array}{ccc} \dot{D} & = & \frac{\beta }{T_{R}}C\\ \dot{R} & = & \frac{\left(1-\alpha \right)}{T_{L}}I+\frac{1-\beta }{T_{R}}C \end{array}$$

It is relevant to note that in (Equation 1), we allow deaths to occur from the unknown infected cases *I*, but that we assume the rate is the same as that for *C*, as well as allowing these deaths to be known. This is intended to reflect circumstances where a person is diagnosed posthumously, or extremely close to death. Meanwhile, in (Equation 5), we assume each fatality to occur in a case where the individual is first diagnosed. While (Equation 1) makes some stronger assumptions, it will soon be apparent that these or similar assumptions are necessary for identifiability. We thus need this for the parameters of the model to be capable of estimation from known data.

The third case is a slight modification of (Equation 1), now allowing *C* and *I* to have two distinct death rates, referred to as $\beta_{1}$ and $\beta_{2}$, respectively. Explicitly, the changed equations are
(6)$$\begin{array}{ccc} \dot{D} & = & \frac{\beta_{2}\left(1-\alpha \right)}{T_{L}}I+\frac{\beta_{1}}{T_{R}}C\\ \dot{R} & = & \frac{\left(1-\beta_{2}\right)\left(1-\alpha \right)}{T_{L}}I+\frac{1-\beta_{1}}{T_{R}}C \end{array}$$

Finally, we consider the case of (Equation 6) but with the death information recorded separately, assigning, $D_{1}$ to *C* and $D_{2}$ to *I*. Here, we consider both to be known as the case where only $D_{1}$ is known was found to have only $\beta_{1}$ identifiable and the case where only $D_{2}$ is known is unreasonable, as confirmed-case-induced deaths are naturally expected to be known.
(7)$$\begin{array}{ccc} {\dot{D}}_{1} & = & \frac{\beta_{1}}{T_{R}}C\\ {\dot{D}}_{2} & = & \frac{\beta_{2}\left(1-\alpha \right)}{T_{L}}I\\ \dot{R} & = & \frac{\left(1-\beta_{2}\right)\left(1-\alpha \right)}{T_{L}}I+\frac{1-\beta_{1}}{T_{R}}C \end{array}$$

Each of the models, with varying assumptions on the assumed known parameters, were run through the SIAN code. In each case there were no instances of parameters which were locally, but not globally identifiable. We also note that in cases where C(t) was the only known variable, α, β, and $R_{0}$ were all not identifiable.

In each case, R(0) was not identifiable, due to the fact that the current value for *R* did not affect the rate at which any compartment was changing. It can be estimated though, if each of the other variables’ initial value is estimated and the total population is known. We see from Table 1 that under even the worst cases, $T_{L}$ and $T_{R}$ may be estimated, which aligns with the common assumption that they can be more straightforwardly inferred from known incidence data. To estimate any other parameter requires either knowledge of the death rate for *I*, or for *I* and *C* to be assumed to have the same death rate. In the cases where the identifiability of parameters failed, not even local identifiability held, meaning that an infinite number of parameter combinations could give rise to the exact same solution for the known variables. Thus, in order to numerically estimate the parameters we moved forward with model (Equation 1), as it had the best conditions for parameter estimation, while still having reasonable underlying assumptions. A similar approach would also work for models with hospitalization data. There, hospitalizations take on a similar role to the deaths in providing indirect information on the unknown case numbers.

As an additional remark, the properties of model (Equation 5) are worse than those of (Equation 1) because for the former the time derivative of *D* essentially provides information for *C* (which is known) and hence does not assist towards identifying *I*. The latter identification is, apparently, possible within (Equation 1). In the case of (Equation 6), $\beta_{2}$ is needed to allow for identifiability (once again to allow for the detection of *I*), while $\beta_{1}$ connected with the confirmed cases does not suffice. It also may not be reasonable to know $D_{1}$ and $D_{2}$ separately as generally, only total death counts are in the reported information. The combination of these factors, as well as the knowledge that even with both death counts, the identifiability only holds in this model when $\beta_{2}$ is known, which is unreasonable if it is assumed distinct from $\beta_{1}$, supports that model (Equation 6) is not as useful practically.

While having two distinct values of β would be more realistic, we see that it creates fundamental issues with the identifiability of the model. We thus made the assumption of a single β value here, in order to use it as a test case for our network. There may be potential methods to resolve these issues while retaining identifiability, but in the present work our intention was to keep the assumptions of the model relatively simple while testing the novel loss function introduced in Section 2.5. The application of the network to a more complex model is left to future work.

### 2.4. Data Processing—Denoising of Data Using a Wavelet Transform

Before feeding our data to the neural network we constructed, we first processed the data using a wavelet transform to separate the noise of the signal from its primary features. Our usage of this process was motivated by many other works where the denoising of an input signal for a neural network using a wavelet transform produced notable improvements [37,38,39]. We implemented code from the PyWavelets package to perform our wavelet analysis and denoising on our signal [40]. We considered the following framework for our data: in our model, each piece of measured signal data can be mathematically expressed as
(8)$$f\left(t\right)=f_{true}\left(t\right)+\varepsilon ,\;\;\;\;\;\forall t\in \left[0,T\right]$$
where $f_{true}\in R^{T}$ is the true data that would have been obtained in ideal measuring conditions and ε comprises the adverse effects of the local environment or faulty activity for the feature, and is referred to as “noise”. To avoid dealing with the noise in the data set, we applied a wavelet transform to denoise the data.

Wavelet theory provides a mathematical tool for hierarchically decomposing signals and, hence, constitutes an elegant technique for representing signals at multiple levels of detail [41]. In general, the wavelet transform is generated by the choice of a single “mother” wavelet ψ(t). In our implementation we used “Symlets 5” as our choice of mother wavelet (details can be found in the code documentation of [40]). Wavelets at different locations and spatial scales are formed by translating and scaling the mother wavelet. The translation and dilation of the mother wavelet are written with the operator U(u,v) acting as follows [42]:$$U_{u,v}\psi \left(t\right)=e^{-u/2}\psi \left(e^{-u}t-v\right)$$

Essentially, we can treat the “daughter” wavelets above, generated by rescaling and shifting the “mother” wavelet, as analogous to the sinusoidal functions of the Fourier transform. The advantage is that the rescaling and shifting allow the wavelets to capture more local behavior. This is performed by taking wavelets with small scales *u*, and shifting across the considered time range using varying values of *v*. A potentially very robust analysis of signals may be conducted in this way, though for our purposes the goal was to use the wavelets to decompose the signal into high and low frequency components, then discard the especially high frequency components as noise.

Now, for a given signal *f*, the wavelet transform of *f*, $\Phi_{\psi }f\left(u,v\right)$, at scale *u* and location *v*, is given by the inner product:$$\Phi_{\psi }f\left(u,v\right)=\int_{-\infty }^{\infty }f\left(t\right){\left(U_{u,v}\psi \left(t\right)\right)}^{*}\;dt=<f,U\left(u,v\right)\psi >$$

Here, star represents the complex conjugate. This is known as the continuous wavelet transform or CWT. To ensure that the inverse CWT is well-defined, we need the following inequality
(9)$$C_{\psi }:=\int_{0}^{\infty }\frac{|\hat{\psi }{\left(\xi \right)|}^{2}}{\xi }d\xi <\infty$$

Here, $\hat{\psi }\left(\xi \right)$ is simply the Fourier transform of ψ. This inequality is referred to as the admissibility condition [42]. One interpretation of this is that the choice of a mother wavelet must have no zero frequency component, i.e., no nonzero constant component. The finiteness of this integral guarantees that the result of the CWT always has a finite $L^{2}$ norm. Generally, once this integral is verified to be finite, the mother wavelet is rescaled by it so that the inverse CWT may simply be defined as
$$f\left(t\right)=\int_{-\infty }^{\infty }\int_{-\infty }^{\infty }\Phi_{\psi }f\left(u,v\right)U\left(u,v\right)\left(\psi \left(t\right)\right)\;dudv.$$

This method of constructing the wavelet transform proceeds by producing the wavelets directly in the signal domain, through scaling and translation. When the signal frequency is higher, the wavelet base with a higher time domain resolution and a lower frequency resolution is used for the analysis. Conversely, when the signal frequency is lower, a lower time domain resolution and higher frequency resolution are used. This adaptive resolution analysis performance of the wavelet transform can effectively distinguish the local characteristics of the signal and the high-frequency noise, and accordingly perform the noise filtering of the signal.

The general steps of the wavelet transform threshold filtering method are as follows:(1)Perform the multiscale wavelet decomposition on noisy time series signals; this process can be continued until the “noisy” or detailed component of the signal is sufficiently low in variance (see Section 3.2);(2)Determine a reasonable cutoff threshold and eliminate the high-frequency coefficients at each scale after decomposition;(3)Perform a wavelet signal reconstruction from the wavelet coefficients after the zeroing process to obtain the filtered denoised signal.

The original signal *f* can be decomposed into $f=A_{1}+D_{1}$ with $A_{1}$ being a lower frequency approximation and $D_{1}$ a high frequency signal. Essentially we are decomposing the original signal into a sum of wavelet terms of the form $U_{u,v}\psi \left(t\right)$, with gradually decreasing values of *u* and suitable values of *v*. After reaching a cutoff for *u*, we reassemble the function using the approximation via lower-frequency and larger-scale wavelets, giving $A_{1}$, and the remaining portion of the function is represented as $D_{1}$. The process may be repeated, treating $A_{1}$ as the new “original” signal, to get the decomposition $A_{1}=A_{2}+D_{2}$. The process may be repeated for several iterations.

In addition to the removal of noise, filtering also smooths the data out as well. This smoothing does not substantially change the actual numerical values or trends, but it does help achieve a better fit for the model. As an ODE, the SICRD model assumes each of the variables is at least continuously differentiable, thus smoothing the inherently nondifferentiable accessible data helps the network perform an analysis on it. This decomposition allows us to obtain a denoised signal, while still retaining information about the exact nature of the filtered noise. With a method to appropriately process a noisy signal and prepare it as a training set, we can now define the network intended to analyze our data.

### 2.5. Setup of the Neural Network

The ability of neural networks to act as universal approximators is well known [43] but the ability to perform accurate estimations in reasonable time frames is a central element of their increasing appeal. A variety of network structures have been developed to suit the features of particular problems. For example, recurrent neural networks, such as the long short-term memory (LSTM) network, have been used to improve pattern recognition in data that have historical dependencies [44].

For our approach, we implemented a new “modified” “Physics Informed Neural Network” (PINN) to learn the values of the parameters and the unknown variable I(t) in our model Equation 1. The PINN approach involves the changing of the loss function for the network, rather than any particular change in the network architecture itself, and in fact is compatible with a wide variety of possible architectures [13,22]. The concept of the PINN is to incorporate some physical law that must be obeyed by the system, and to introduce an extra term into the loss function which becomes smaller the closer the network output adheres to the law [27]. The approach was developed to estimate parameters and do forecasting in cases where data are only available at relatively sparse time intervals. The method is also robust to issues of overfitting, making it particularly useful for studies in limited-information regimes such as disease modeling [27]. While the method has been applied to more complex network architectures [22], we considered only the fully connected feed-forward network in our work as we tested our “proof of concept” modification.

Initially, this concept was introduced in the context of nonlinear partial differential equations, though we see that our case of a nonlinear ordinary differential equation still falls into the applicable category (and other studies have also indeed used them in this context [28,45]). The common approach is to simply take the difference of the differential equations’ sides and introduce that as an extra term to minimize in the loss function.

This is sufficient for many applications but we found it unable to generate acceptable results in our case. This was largely due to the entirely unknown variables *S*, *I*, and *R* in our model. We also only had a single time series from the system to use in estimating the parameters, whereas in many attempts to estimate system parameters using PINN methods, an entire ensemble of starting conditions and subsequent trajectories is used. For example, the DeepXDE deep learning method was used to attempt to estimate the parameters of a Lorenz system but needed to take a large collection of different starting trajectories to achieve good estimates [46]. Many methods of augmenting the PINN with other approaches exist [22], but generally the core structure of the loss function, the defining feature of the PINN, is not adjusted to improve training. Once a choice of modeling equation is made, it is directly used for the loss terms. The work of [29] is an appealing recent example with a reformulation of the original two-dimensional ODE system with an equivalent one-dimensional second-order ODE. Their goal was to formulate a more straightforward expression for the infection rate parameter, though the inherent effect of the approach on the network training was not considered.

To address our case, we developed a new method, changing the structure of the loss function to that in Equation (Equation 14). The new terms still constituted the “physical law” for our system, but were of a form where the network’s gradient descent was far more stable. This novel approach, in our view, nontrivially extends PINN methods, and hence that is why we refer to it as a “modified” PINN system. This reveals the importance of tailoring the loss function not only to each system, but to the specific circumstances of accessible variables and parameters for a given problem. The tests on artificially generated data in Section 3.1 were extremely promising, with accurate estimations of the unknown parameters $R_{0}$ and α.This was performed with only a single time series of C(t) and D(t) being used to perform the estimation. The network architecture was also the simple fully connected feed-forward network, yet it still performed well. This attests to the efficiency of this suitably chosen loss function.

Before we precisely define our modified approach, we first review the standard PINN implementation. Generally, although many forms of PINN systems have been used, the loss function is basically of the same form [13,22]. To give a precise definition for the ODE case, consider an n-dimensional ODE system defined by
(10)$$\dot{X}=f\left(X,p\right),$$
where f:=f(X,p) is some function of all our variables *X*, and the given system parameters are *p*. The loss term ${\mathrm{Loss}}_{pinn,1}$ may then be defined by
(11)$${\mathrm{Loss}}_{pinn,1}:=\overset{n}{\sum_{i=1}}{\mathrm{MSE}}_{t}\left(f_{i}\left(X_{est},p_{est}\right)-{\left({\dot{X}}_{est}\right)}_{i}\right).$$
where $\left(X_{est},p_{est}\right)$ are the estimated values of the variables and parameters output from the neural network. ${\mathrm{MSE}}_{t}$ is the mean square error up until time *t*. This loss essentially measures how well the output of the neural network is obeying the physical laws that govern the system. Note that the estimated values of the system parameters appear in *f* and thus may be learned using this loss term.

The PINN loss is then combined with the standard loss function, with $X_{test}$ being the given test data:(12)$${\mathrm{Loss}}_{mse,1}=\overset{n}{\sum_{i=1}}{\mathrm{MSE}}_{t}\left({\left(X_{est}\right)}_{i}-{\left(X_{test}\right)}_{i}\right).$$

Thus, we have the total loss:(13)$${\mathrm{Loss}}_{1}={\mathrm{Loss}}_{pinn,1}+{\mathrm{Loss}}_{mse,1}$$

Our initial testing essentially implemented the above scheme using the “Disease Informed Neural Network” code as a base [45]. Our version of ${\mathrm{Loss}}_{mse,1}$ could only include *C* and *D*, due to these being (assumed to be) the only known variables, but otherwise the loss function was the same. It was found that the network consistently led to incorrect values for all variables and parameters. The existence of unknown variables created several complications: the network had a fundamental gap in its training data, and the network was trying to estimate the unknown variables (S,I,R) in addition to estimating the unknown parameters. The presence of so much unknown information created substantial problems in the network’s ability to minimize loss for any particular term from $f_{i}$, without creating significant losses in another element of the equations $f_{i}$. The standard PINN loss terms also varied quite substantially in magnitude. The size of $\dot{D}$ was much smaller than $\dot{C}$, and thus might impact the network’s learning less substantially, despite how crucial the death data were (as seen in our analysis in Section 2.3).

Motivated by these deficiencies, we developed a modified interpretation of the PINN concept in the following way. If our variables for an ODE system are given by the vector X(t), we choose functions $g_{i}\left(X_{est},p\right)$ (where we recall *p* is the vector of parameters for the system) which should each be 0 if the time series $X_{est}$ precisely obeys the ODE system. Concretely, we consider the SICRD model and derive the following functions $g_{i}$ from Equation (Equation 1):(14)$$\begin{array}{ccc} g_{1} & = & \left(1-\alpha \right)\dot{C}-\alpha \dot{D}/\beta +C/T_{R}\\ g_{2} & = & \frac{\alpha I}{T_{L}}-\dot{C}-\frac{C}{T_{R}}\\ g_{3} & = & \dot{I}+\dot{S}+\dot{C}+\dot{D}/\beta \\ g_{4} & = & R_{0}IS-NT_{L}\dot{S}. \end{array}$$

Notice that these expressions involve suitable modifications of the dynamical equations through (linear) combinations thereof. We can then use these functions to define the term:(15)$${\mathrm{Loss}}_{pinn}=\sum_{i=1}^{i=4}\mathrm{MSE}\left(g_{i}\right).$$

Then, the total loss function is given by
(16)$${\mathrm{Loss}}_{total}={\mathrm{Loss}}_{mse}+{\mathrm{Loss}}_{pinn},$$
where
(17)$${\mathrm{Loss}}_{mse}=MSE\left(C_{est}-C_{test}\right)+MSE\left(D_{est}-D_{test}\right).$$

Some issues with the performance of the code were found when trying to use ratios or log scales to try to equalize the terms in the MSE loss, but ultimately the performance of the network worked well enough with the standard form. Alternative approaches with these methods of equalizing could be considered though. With these new loss functions, the network converged quickly and accurately when tested on artificially generated data (results in Section 3.1).

Part of the motivation for this choice is that it excludes from consideration the recovered population for which there are always unidentifiable features (such as R(0)) and focuses on the rest of the populations, including an effective rewrite of the conservation law associated with the total population. We also note that in our new equations, we tried to reduce how frequently multiple unknown quantities appeared together. To illustrate this point, $g_{1}$ involves only known quantities except for α, if we assume β to be known. This allows this equation to be used to learn one parameter. If α is in principle known from $g_{1}$, then all quantities except *I* are known in $g_{2}$, and thus *I* can be well estimated. Then, all quantities except $\dot{S}$ are in principle known in $g_{3}$, letting $\dot{S}$ and *S* themselves be estimated. Finally $g_{4}$ can then estimate the basic reproduction number $R_{0}$.

Practically, this is not precisely what is happening, as each term is being minimized simultaneously by the network. This is just meant to illustrate in principle how this structure reduces the interference of multiple competing unknown quantities (a detail known to cause issues in the training of some networks [47]). We also note that all of the equations are of similar order, whereas in the standard PINN network, the quantity $\dot{D}$ is substantially smaller than any other derivatives appearing in the system. In that light, the present restructuring of the equations systematically builds the optimization of the system parameters.

This approach is in no way unique to our particular choice of system either. It shows that the concept of the PINN can be made broader than the standard choice of how to incorporate information from the original set of differential equations a system is assumed to obey. Various different ODE or PDE systems could have alternatively derived formulations that can then be used to train a neural network. The most crucial point of our example is that it shows the existence of a case where the standard PINN approach is insufficient, but the modified approach is extremely effective.

As far as the structure of the network itself is concerned, we note that it is a simple fully connected neural network, constructed with some minor modifications of the base code of [45]. The hidden dimension and learning rate initially presented in [45] created some issues, due to the original work assuming that all of the population compartments were known. This assumption allowed the original code to still converge reasonably well with a much lower learning rate, as well as a lower hidden dimension. In our implementation, we created 6 layers for our fully connected linear neural network, with hidden dimension 64. The nonlinear activation function in each layer was chosen to be the usual ReLu function, defined as: ReLu(x)=x, if x>0, and ReLu(x)=0 otherwise. The learning rate was taken to be lr=0.0001, while the optimizer used was Adam [48].

## 3. Results

### 3.1. Artificial Data Testing

In order to test the validity of our estimation method, we first tested it on a set of artificially generated data. With artificial data, we can know the “hidden” variables and parameters while the network is only fed the corresponding information known in the real case. Then, the network’s resulting estimates can be compared with the true values to evaluate its performance, unlike with real data where this information is unknown. We assumed the real system corresponded at least roughly to the description of (Equation 8), behaving on average as the ODE system (Equation 1) prescribed, with some random noise.

We tested the neural network on artificial data with no noise, i.e., the ideal case, to verify its baseline capabilities. We generated the time series X(t), our vector of variables for the SICRD model, using a standard ODE solver. We then fed only the “known” variables C(t) and D(t) to our network with our new loss function (Equation 17), to generate an estimated time series and parameter values. We could then measure ${\mathrm{MSE}}_{t}\left(I_{est},I_{test}\right)$, as well as the difference between the estimated parameters and the true values. Small MSE values suggested that so long as our base assumptions about the system were reasonable, our method accurately estimated the infected population and unknown parameters.

For our testing, we fixed the known values $T_{L}=8.3$, $T_{R}=9.2$, and β=0.05. $R_{0}=2$ and α=0.8 were chosen for the unknown parameter values. We ran over a time interval of 40 days with 400 data points distributed evenly. The code was run for 50,000 epochs and executed in 3 h. We see in Figure 2 that the correct values for the unknown parameters were converged to quite rapidly and in a stable fashion. The oscillations in the loss function were not unexpected, as we noted that the oscillations were small due to the logarithmic scale. This was essentially just the process where once the loss was small enough, the network could no further diminish the loss without passing through regions where the loss increased temporarily while the Adam optimizer ran.

Moreover, the total MSE loss along the time series we obtained was:Stotalloss:0.09977;Itotalloss:0.00069;
Dtotalloss:0.00262;Ctotalloss:0.00093

Visually, we can see in Figure 3 how each learned time series (the dotted lines) overlapped almost precisely with the true time series. Clearly, in this example, the modified PINN performed quite sufficiently in identifying both the parameters and the unavailable time series information.

We also present the results from the standard PINN, i.e., directly using the form from Equation (Equation 11) for the PINN loss. Testing was performed under the same conditions as for the modified PINN, with the results presented in Figure 4 and Figure 5. We note with an initial value of α as 0.82, as we did in the modified PINN, α decreased directly to 0.7 during the training, as seen in the first graph. Moreover, the fitted values for *C* and *D* were quite poor, as seen in Figure 5 despite these being known quantities. Without modification, the PINN gave results which were not merely poor, or took substantial time to converge but the results also consistently converged to entirely incorrect values. Our modification was *crucial* towards recovering accurate estimates of *I*, α, and $R_{0}$.

### 3.2. Wavelet Denoising Results

With the testing of the network on artificial data concluded, the next step involved the application to real data sets. The given time series, C(t) and D(t), each had noticeable noise. We applied the wavelet decomposition defined in Section 2.4 before applying the network to the filtered signal in Section 3.3.

We demonstrate several layers of wavelet decomposition from the original signal, illustrated in Figure 6. Note that the overall shape and scale of the signal was preserved in this smoothing process. $A_{i}$ for higher values contained less of the precise information, but that information was preserved in the corresponding sequence of $D_{i}$. A further analysis which incorporated the signal information for $D_{i}$ could have been performed, but we did not pursue that avenue.

The $A_{1}$ approximation was used for the training of our network, as this gave a reasonable balance between being smoothed out well enough for ease of training, while still retaining substantial details of the original sequence.

### 3.3. Real Data Testing

After verifying the effectiveness of the network at testing on artificial data and processing via the wavelet transform, the next step was to perform testing on real data. Only *C* and *D* were known variables with the real data though, so it was not possible to evaluate the network based on its ability to estimate the unknown variables. It was still possible to at least measure how well the network was able to fit the available data while generating estimates for the unknown variables and parameters. The network was trained on reported data from Alabama for 90 days starting on 1 May 2020, with the resulting estimation the network produced for *C*, *D*, and *I* given in Figure 7.

The estimated values for the assumed unknown parameters were $R_{0}=1.8162$, β=0.0098, and α=0.8762. The values $T_{L}=8.3$ and $T_{R}=9.2$ were assumed based on medical record results [21].

We see that the network was able to generate very close fits to the known variables *C* and *D* over the 90-day time span, despite the relatively noisy data. We did not train over longer intervals of time as the assumption of constant parameter values became more unrealistic. A potential approach using time-varying parameters is tested in Section 3.3.1, whereas here, testing was constrained to constant parameter values in order to perform our initial verification.

Thus, so long as the model had a reasonable fit to the assumption of Equation (Equation 8) with our SICRD model, the estimates for I(t) and the unknown parameters should be relatively accurate in their determination. Note that the general modification concept could easily be applied to other models as well, though new functions serving the role of the equations in (Equation 14) would have to be derived.

#### 3.3.1. Time-Dependent Parameter Estimation

Our next step was to perform an estimation using the time-dependent parameter model of Section 2.1. We found that our modified PINN was able to effectively and efficiently minimize loss over a scale of about 60–90 days. We present an example estimation of varying values for the parameters using available data in Alabama from 1 May 2020 to 31 December 2020, with a rolling window of 90 days, using the approach described in Section 2.1.

We present our estimates of the time-dependent parameters in Figure 8 and Figure 9. If the true values were constant, or close to constant, we would not expect to see such substantial variation in the graphs, so it seems that realistically the parameters should be assumed to vary for long-enough timescales. The value of $R_{0}$ seems relatively stable, which may suggest that there were no substantial social distancing efforts implemented. Meanwhile, the variation of the confirmation rate α seems to indicate a difficulty in keeping testing up with infections at first, though some eventual adaptation takes place. The highly “noisy” look of α may be due to artifacts of how confirmed cases are reported. The most substantial variation though is the drop in β, which may reflect improvements in the treatment of severe cases of COVID-19. Care has to be taken in interpreting the values though, as uncertainties in the measured information and accuracy of the model are always present.

The results are suggestive of the viability of the use of this network approach on systems with time-varying parameters. In the following section, we restrict ourselves to a 90-day time window with the assumption of fixed parameters to make estimations on all 50 U.S. states more feasible.

#### 3.3.2. Ranking States by Testing

Now that we have a network that is able to, with reasonable effectiveness and efficiency, estimate these unknown quantities, we consider a concrete question this information can help us answer: “Which states have the worst relative rates of testing?”. This is an important piece of information for federal policymakers, as it determines which areas would be in the most dire need of additional federal aid in the form of (limited) testing supplies. The sense of “worst relative testing rate” can be quantified by finding which states have the highest ratio of *I* to *C*, averaged over a given time interval. Explicitly, we selected our corresponding diagnostic to be:(18)$$A\left(t\right)=\frac{1}{t}\sum_{k=1}^{t}\frac{I(k)}{C(k)}$$

The importance of A(t) is that it lets us estimate which areas of the country are struggling the most to efficiently test their populations, relative to the prevalence of the disease. The idea of this particular metric is that it gives more specific information than just simply taking the estimation for the unknown infected population. Taking the infected population alone might just give information on locations that happen to have an exceptionally high number of infections occurring. A(t) more directly identifies areas (in our case: states) where information is scarce relative to current levels of infection. This could be vital for identifying population centers that may not have a high number of infections currently, but are highly vulnerable to potential outbreaks due to poor testing infrastructure. The information from our direct estimates for I(t) as well as for A(t) together give a more robust measure of the current testing situation.

The importance of early intervention in the spreading of COVID-19 is already known [21], so the potential for this metric to provide an early warning is highly valuable. Even with limitations in estimating I(t), as long as the network is able to distinguish population centers of major need from ones of minor need, then it is still useful as a tool for policy-making.

The confirmed case counts over 90 days starting from 1 May 2020, for the states with the top 3 and bottom 3 estimated A(t) values are given in Figure 10. The case counts over 6 months starting from same date are given in Figure 11. The values of A(t) over a 90-day interval starting on 1 May 2020, estimated by our network, are given in Table 2 and Table 3, as well as the estimated values for the unknown parameters. We see that the estimated value for A(90) did not correlate in a direct and simple way with the values of the parameters. Thus, the metric did appear to provide extra information that could not be extrapolated simply from the parameter estimates alone.

At the highest extreme, Idaho had a roughly 2.8 to 1 ratio for infected to confirmed. We note that while the time frame for this type of statistics was early in the pandemic, a state’s ranking may be suggestive towards subsequent effects. During the tested period, Idaho neglected to implement mask mandates in the summer of 2020 [49], later facing extreme overburden of its medical system by December 2020 [50], and further on having the lowest vaccination level later in September 2021 [51], features that appeared to be in line with our findings (although, of course, a further exploration of such case examples is useful to consider in future efforts). Pennsylvania, on the other hand, had the lowest ranking, with a ratio of 1.011, just barely more cases of unknown infections estimated than confirmed cases. Pennsylvania had already instituted some mask mandates in April 2020 [52], and these mandates were substantially expanded in July 2020 [53]. Universities within the state also began declaring that classes would be moved online for the fall semester [54], with all of these features indicating a higher degree of preparedness to address the pandemic. This once again suggests the relevance of further consideration of this type of diagnostic.

Overall, the existence of such a substantial disparity between states supports that this type of analysis may be interesting to consider and explore further. While we do not intend to reach any definitive conclusions in this connection since, realistically, there are many factors that may interfere with an accurate picture of this estimate, the consideration of the case examples that we have examined suggests that a further examination of such ideas and diagnostics would be worthwhile.

## 4. Discussion

In the present work, we chose a modified version of the SIR model for the time evolution of COVID-19, motivated by the features of the disease and the nature of available data. The model allowed the distinguishing of confirmed and unconfirmed cases, as well as a recording of the number of fatalities due to the disease. Upon verification of the identifiability of the unknown quantities within the model (based on the practically available data), we were able to use our modified PINN approach to analyze available data, with a special loss function tailored to the lack of available information.

The most striking result was the substantial improvement that our modification made, as it was not obvious that a reformulation of the governing equations should have such an impact on the network’s learning. In principle the equations of (Equation 14) and (Equation 1) with N=S+I+C+D+R contain the same information. The exact nature of this dependence has likely gone unnoticed due to the basic implementation of the PINN being sufficient for most applications. This kind of circumstance though, performing inference when some variables are unknown, is not a rare one. The PINN was originally developed to be effective for inference with sparse data [27] but our work demonstrates that it extends to even incomplete data effectively, when appropriately modified.

There are, of course, many further refinements that could be made to our methods. The SICRD model could always be further separated into more distinct population compartments relevant to the spread of the disease. Compartments for exposed, though not yet infectious individuals, explicitly asymptomatic individuals, hospitalized individuals, and more could be included, and we discussed some principles on the basis of which such considerations could be explored. Additional topics that are quite relevant are the consideration of aspects of age stratification [15,18,55], as well as the incorporation of the role of vaccines and immunity waning (for later time frames than the ones considered herein, during which a vaccine was available) [9,56,57,58]. We could also consider cases where confirmed individuals may still infect others, albeit at a reduced rate.

Our current framework, however, was intended to be used to study the development of the disease on relatively short time frames in the early pandemic stages, mitigating the effect of immunity loss as individuals would most likely still have immunity. Within this time frame, we could also disregard vaccinations, though, as stated earlier, the inclusion of vaccination effects could be incorporated if later times were considered. Instances of confirmed individuals spreading infection were assumed to be an exception as most individuals followed appropriate quarantining procedures on a positive test result.

The crucial role of the death data (or hospitalization data in an SEAHIR model [16]) for estimation does raise the important issue of the validity of such data. The existence of underreporting, delays, and misattribution (namely, the important concern about “death from COVID” vs. “death with COVID” [59]) can create potential issues that warrant further examination. Nevertheless, we consider it to be the most relevant starting point for the study conducted herein and certainly a far more reliable one than the total case count, as has been explained also earlier in connection to studies from the CDC; see [18] for a relevant discussion.

Indeed, it is also relevant to point out that, as demonstrated in Section 2.3, the unknown infected population is impossible to estimate from recorded case counts alone. Hospitalization case information has also been incorporated into other methods of disease modeling with effective results so there is an existing precedent [16]. Such issues exist in models of waterborne illness, where the presence of difficult-to-track pathogens in water creates similar complications to our asymptomatic cases [34]. Further work can and should include further data to improve the model’s robustness. What the present work does is provide a mathematical basis and a computational implementation such that, even with a more complicated model, the PINN approach will still be reasonably applicable. Before our modification, it was not clear that a PINN could feasibly work even with our simpler model, in the presence of multiple unknown variables.

Ultimately the simplifications made for the model, as well as for the architecture of the neural network, were taken so that the evaluation of the novel loss function could be performed with minimal complications. Future work could naturally focus on expanding both the model’s complexity as mentioned above as well as incorporating more nuanced network structures. The work of [25] on graph neural networks gives a very promising direction for future work in both directions, with a natural extension of the model to incorporate human mobility, paired with the natural choice of a graph neural network architecture. Indeed, the study of such metapopulation models of wide appeal within the modeling of COVID-19 [4,21], in conjunction with some of the technical approaches and methodologies presented herein, constitutes a promising direction for future study.

The approach could also be suitably adapted to some form of recurrent neural network (RNN) as well. RNNs have already been used in disease modeling [26] and the combination of the LSTM, a specific form of RNN, with the PINN concept has been attempted in other works with notable success [22]. A limitation of our current work is the inability to estimate more of the unknown parameters simultaneously. The modified PINN still struggles to perform well if it must estimate parameters $T_{L}$ and $T_{R}$ in addition to $R_{0},\beta ,$ and α. There are simply too many simultaneous unknowns for the network to perform estimation efficiently. Indeed, this issue of numerous local minima that perform equally well has been encountered in various other studies; see, e.g., [18]. An RNN approach could be efficient enough to allow this broader parameter estimation.

A particularly relevant direction, in terms of how widely it may expand our results, would be a rigorously proven criterion for alternate PINN loss terms. We gave an informed explanation of why the choice of Equation (Equation 14) led to better performance, but without a formalized proof. A natural future goal could be to formally prove that an alternate set of loss terms with fewer unknowns generally leads to faster or more accurate convergence. Some analysis on this concept of a network’s difficulty to balance multiple competing objectives does exist [47], so it would be reasonable to expect a form of generalized criteria to hold. If such a proof could be made, it would allow our approach to be applied extremely broadly with a general recipe for how to adapt it to a particular problem. It should also be added the very recently proposed idea of incorporating causality into the loss function of the PINNs [60] may also be quite relevant to consider in the present context. In addition, one could extend the study using graph neural networks [61].

## Figures and Tables

**Figure 1 viruses-14-02464-f001:**
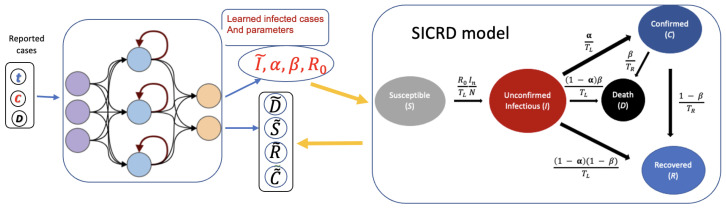
Schematic of the modified-PINN model. We assume here that *I* and the parameters $R_{0},\alpha ,$ and β are not known.

**Figure 2 viruses-14-02464-f002:**
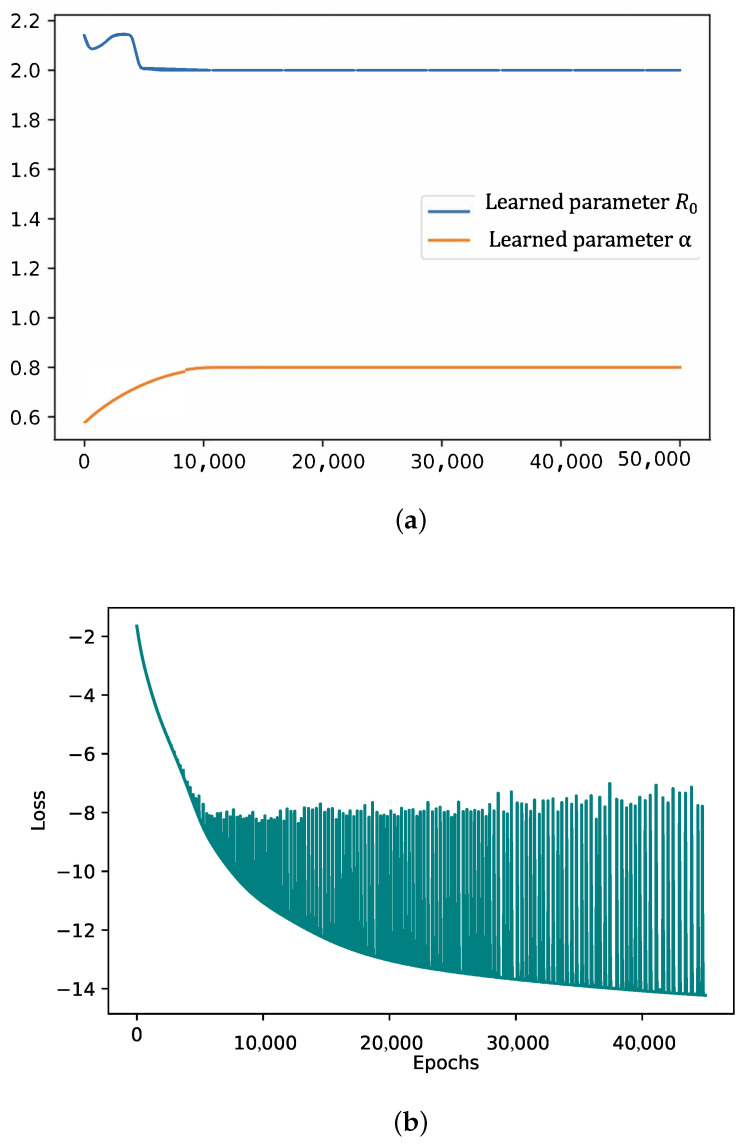
(**a**) The learned parameters $R_{0}=2$ and α=0.8 in the artificial data testing. (**b**) The log-loss curve for the training process as a function of the number of Epochs.

**Figure 3 viruses-14-02464-f003:**
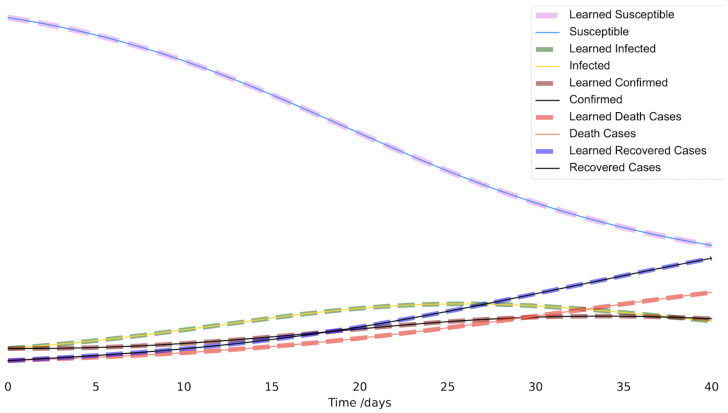
Artificial data time series compared with the network estimated values for each quantity; see the legend for each of the relevant comparisons. Note that the network only had access to the data for *C* and *D*. The true curves for I,R, and *S* were unknown to the network.

**Figure 4 viruses-14-02464-f004:**
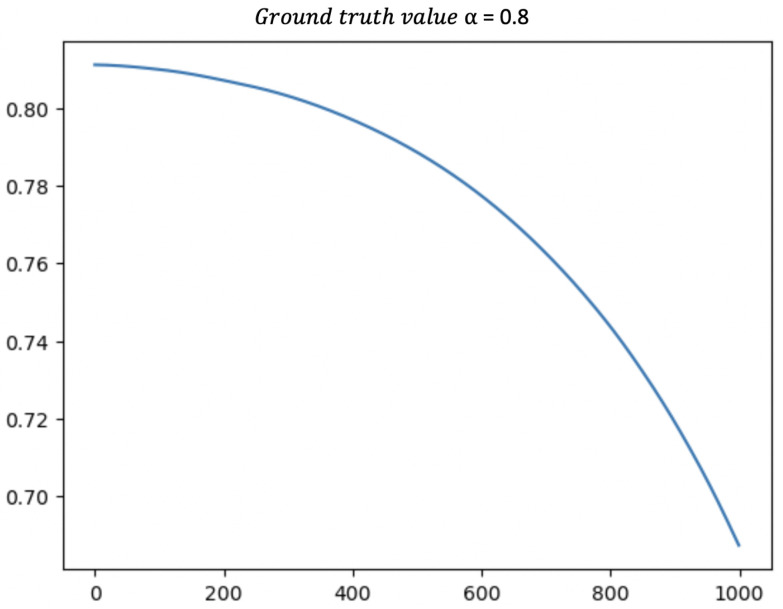
The learned parameter α (with real value α=0.8) using the standard PINN loss function.

**Figure 5 viruses-14-02464-f005:**
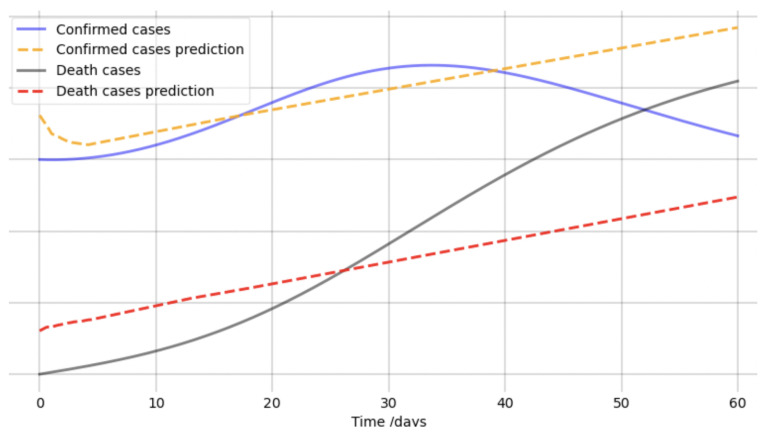
Estimated *C* and *D* using the standard PINN loss function.

**Figure 6 viruses-14-02464-f006:**
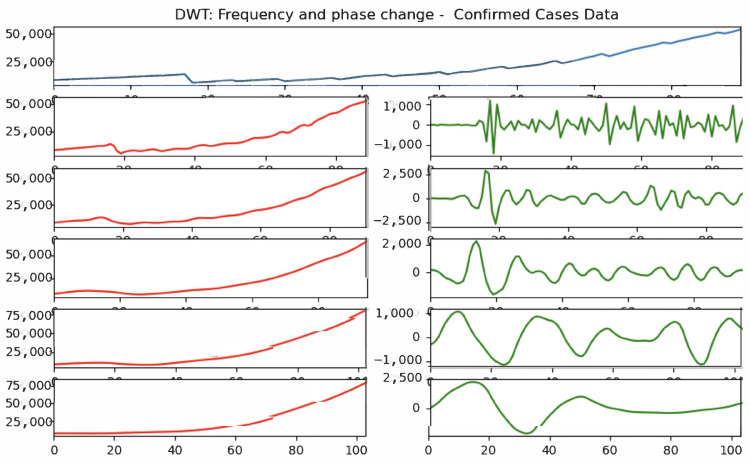
Wavelet decomposition of confirmed case data for Alabama starting 1 May 2020. The original time series and the subsequent five layers of decomposition into approximations ($A_{i}$) on the left and high-frequency signal ($D_{i}$) on the right are shown.

**Figure 7 viruses-14-02464-f007:**
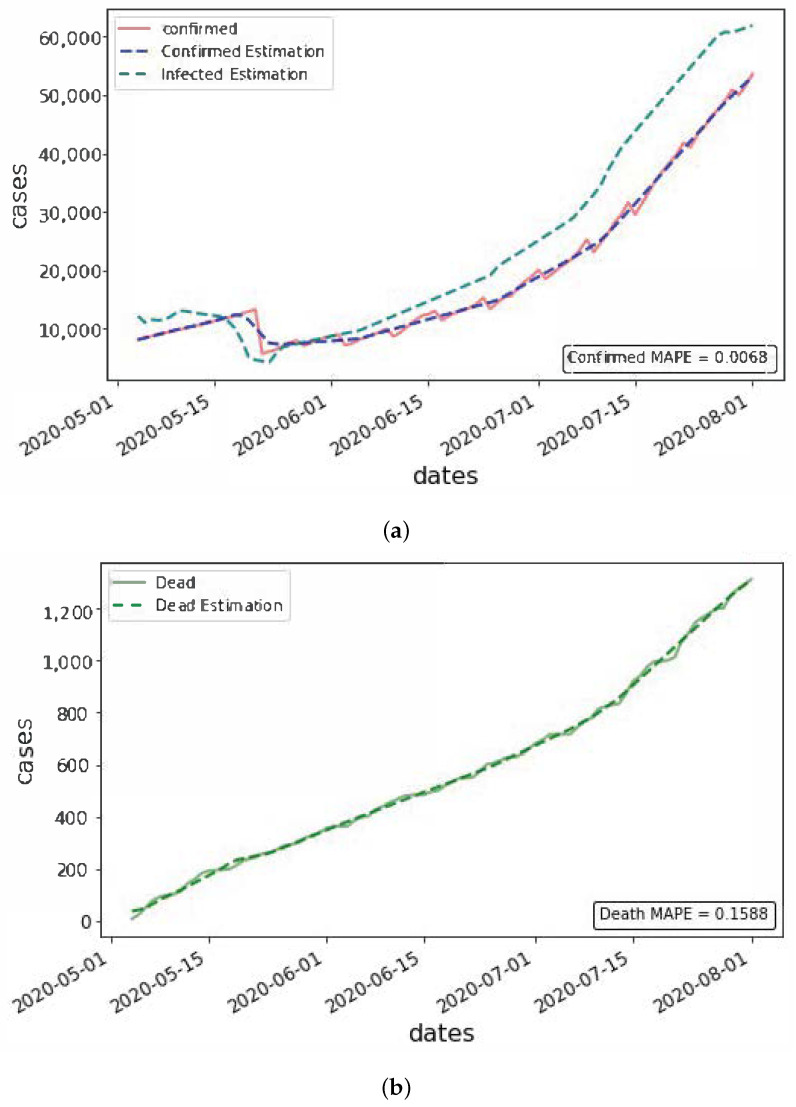
(**a**) The estimated value for infected cases, with corresponding fitted values for confirmed cases, compared with the actual confirmed count. (**b**) The fitted values for death cases, compared with the actual death count. In each case the mean absolute percentage error (MAPE) is included.

**Figure 8 viruses-14-02464-f008:**
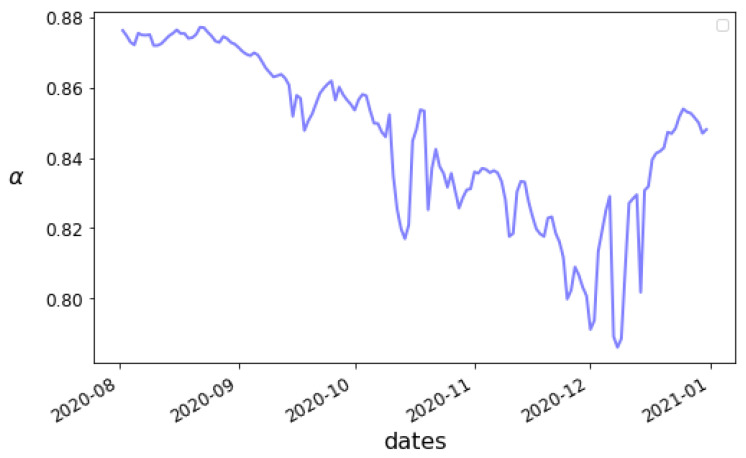
Learned values of the time-dependent parameters α and β as a function of time from 1 May 2020 for Alabama.

**Figure 9 viruses-14-02464-f009:**
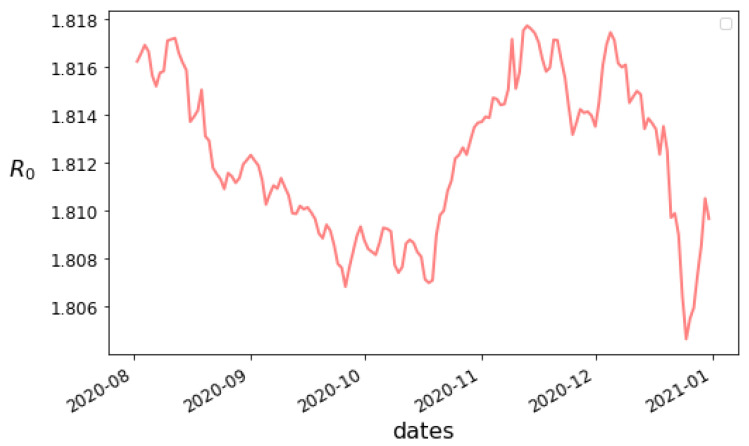
Learned values of the time-dependent parameter $R_{0}$ as a function of time from 1 May 2020 for Alabama.

**Figure 10 viruses-14-02464-f010:**
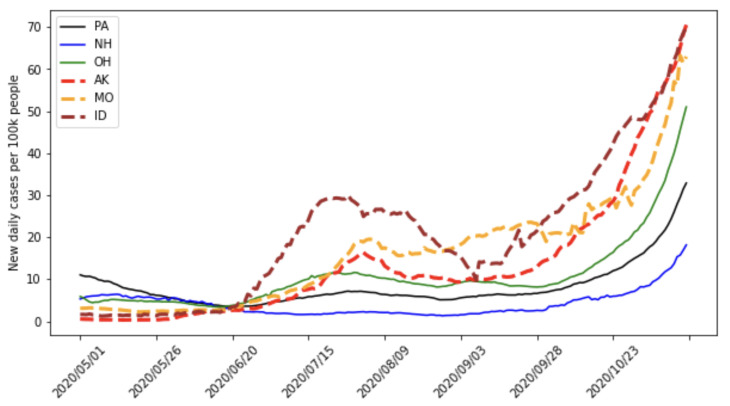
For 90 days starting from 1 May 2020, we plotted the confirmed cases of the bottom 3 estimated A(t) values, corresponding to states PA, NH, and OH, as well as the top 3 estimated A(t) values, corresponding to states ID, AK, and MO. We can see that ID, AK, and MO had a smaller number of cases in May 2020. Their confirmed case counts significantly grew after the middle of June though.

**Figure 11 viruses-14-02464-f011:**
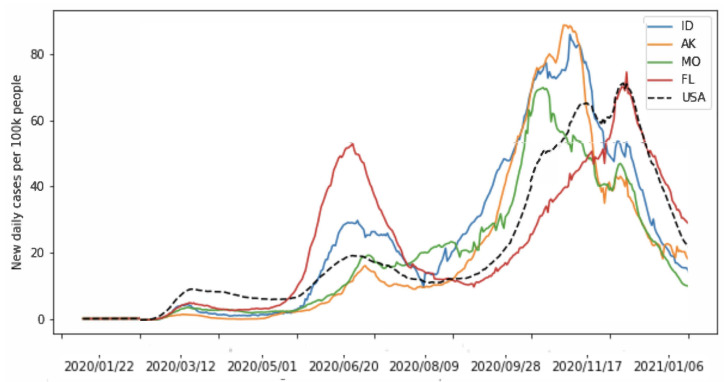
Six months of confirmed cases for these 6 states. We see that the trend for ID, AK, and MO continues for months past the time period over which we estimated A(t).

**Table 1 viruses-14-02464-t001:** Comparing Model Identifiability.

Model	Known Variables	Known Parameters	Globally Identifiable	Not Globally Identifiable
(Equation 5)	C(t),D(t)	$T_{L},T_{R},\beta$	[C(0),D(0)]	$\left[\alpha ,R_{0},S\left(0\right),I\left(0\right),R\left(0\right)\right]$
(Equation 5)	C(t),D(t)	none	$\left[T_{L},T_{R},\beta ,C\left(0\right),D\left(0\right)\right]$	$\left[\alpha ,R_{0},S\left(0\right),I\left(0\right),R\left(0\right)\right]$
(Equation 1)	C(t),D(t)	none	$\left[T_{L},T_{R},\alpha ,\beta ,R_{0},S\left(0\right),I\left(0\right),C\left(0\right),D\left(0\right)\right]$	R(0)
(Equation 6)	C(t),D(t)	$T_{L},T_{R},\beta_{1}$	[C(0),D(0)]	$\left[I\left(0\right),S\left(0\right),\alpha ,\beta_{2},R_{0},R\left(0\right)\right]$
(Equation 6)	C(t),D(t)	$T_{L},T_{R},\beta_{2}$	$\left[\alpha ,\beta_{1},R_{0},S\left(0\right),I\left(0\right),C\left(0\right),D\left(0\right)\right]$	R(0)
(Equation 6)	C(t),D(t)	$T_{L},T_{R},\beta_{1},\beta_{2}$	$\left[\alpha ,R_{0},S\left(0\right),I\left(0\right),C\left(0\right),D\left(0\right)\right]$	R(0)
(Equation 7)	$C\left(t\right),D_{1}\left(t\right),D_{2}\left(t\right)$	$T_{L},T_{R},\beta_{1}$	$\left[C\left(0\right),D_{1}\left(0\right),D_{2}\left(0\right)\right]$	$\left[\alpha ,\beta_{2},R_{0},S\left(0\right),I\left(0\right),R\left(0\right)\right]$
(Equation 7)	$C\left(t\right),D_{1}\left(t\right),D_{2}\left(t\right)$	$T_{L},T_{R},\beta_{2}$	$\left[\alpha ,\beta_{1},R_{0},S\left(0\right),I\left(0\right),C\left(0\right),D_{1}\left(0\right),D_{2}\left(0\right)\right]$	R(0)
(Equation 7)	$C\left(t\right),D_{1}\left(t\right),D_{2}\left(t\right)$	$T_{L},T_{R},\beta_{1},\beta_{2}$	$\left[\alpha ,R_{0},S\left(0\right),I\left(0\right),C\left(0\right),D_{1}\left(0\right),D_{2}\left(0\right)\right]$	R(0)

**Table 2 viruses-14-02464-t002:** States organized by ranking values estimated over 90 days starting at 1 May 2020.

State	A(90)	$R_{0}$	β	α
Pennsylvania	1.011	1.8139	0.0375	0.8598
New Hampshire	1.0296	1.8125	0.0475	0.83
Ohio	1.0547	1.8093	0.0089	0.9099
New York	1.0549	1.7952	0.0032	0.8898
South Dakota	1.0656	1.8015	0.0237	0.8037
Maine	1.0722	1.8028	0.0238	0.8288
Indiana	1.0768	1.8107	0.0212	0.9268
Massachusetts	1.0771	1.8033	0.0126	0.9134
Nebraska	1.0771	1.8159	0.0069	0.8671
New Jersey	1.0848	1.7946	0.0139	0.8585
Connecticut	1.0895	1.7966	0.0102	0.8884
Rhode Island	1.096	1.7964	0.0091	0.9046
Michigan	1.1105	1.8059	0.0188	0.8416
Delaware	1.1361	1.8026	0.014	0.8804
District of Columbia	1.1662	1.7992	0.0077	0.9101
Iowa	1.1779	1.8039	0.0172	0.8872
Wisconsin	1.2021	1.8038	0.0179	0.8872
Minnesota	1.2078	1.8166	0.0417	0.841
Colorado	1.2115	1.8068	0.0052	0.8836
Maryland	1.232	1.8022	0.006	0.8888
North Dakota	1.2371	1.8085	0.0183	0.8802
New Mexico	1.2607	1.8063	0.0142	0.9054
North Carolina	1.2708	1.8063	0.019	0.9193
Wyoming	1.2844	1.8081	0.0123	0.8875
West Virginia	1.286	1.809	0.0133	0.8939
Virginia	1.3242	1.8127	0.0051	0.8745
Illinois	1.3434	1.8022	0.014	0.8022

**Table 3 viruses-14-02464-t003:** States organized by ranking values estimated over 90 days starting at 1 May 2020.

State	A(90)	$R_{0}$	β	α
Utah	1.3906	1.8144	0.0072	0.8667
Louisiana	1.3956	1.8112	0.0165	0.8968
Vermont	1.4032	1.8264	0.0024	0.7205
Kentucky	1.4166	1.8184	0.0066	0.873
Tennessee	1.4332	1.8156	0.0106	0.8761
Mississippi	1.4338	1.814	0.0259	0.9032
Alabama	1.4714	1.8162	0.0098	0.8762
Oklahoma	1.4979	1.8119	0.0182	0.9009
Arkansas	1.505	1.8058	0.0181	0.9048
Kansas	1.5718	1.8142	0.0017	0.7393
Oregon	1.6298	1.8142	0.0054	0.8221
Washington	1.6314	1.8081	0.0188	0.8081
Nevada	1.6836	1.8097	0.0043	0.7946
Missouri	1.6889	1.8028	0.0069	0.9101
Texas	1.7384	1.8037	0.0201	0.9028
Georgia	1.7721	1.8125	0.0077	0.83
Arizona	1.7958	1.8063	0.0078	0.8814
California	1.8618	1.8618	0.0091	0.8898
South Carolina	1.8715	1.8124	0.0118	0.8866
Hawaii	2.0421	1.8235	0.0054	0.7464
Puerto Rico	2.0655	1.8159	0.0139	0.8037
Florida	2.1034	1.8139	0.0375	0.9134
Montana	2.1703	1.8167	0.0132	0.8808
Alaska	2.7863	1.8165	0.0022	0.7136
Idaho	2.8028	1.8127	0.0036	0.7543

## Data Availability

All nonsimulated data on COVID-19’s development used in this paper were pulled from the tool developed in [33]. The tool aggregates data from a variety of data sources including the WHO, each state’s individual department of health, and the CDC. The data used included reports on case counts, active cases, and deaths within each individual U.S. state. We also include a direct link to the github for the tool here: https://github.com/CSSEGISandData/COVID-19 (accessed on 29 August 2022).

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
