# Peer review of "A Modified PINN Approach for Identifiable Compartmental Models in Epidemiology with Application to COVID-19"

_viruses, 2022, doi:10.3390/v14112464_

Round 1

Reviewer 1 Report

The authors use Physics informed networks using a specific modification of the SIR model in order to analyse COVID data. The analysis is interesting and useful.  On shortcoming of the paper is that it does not refer to the first application of PINNS in COVID data, viz. the paper of Barmparis et al, Quantitative Biology 10, 139 (2022). The authors should compare their approach and results with this paper and analyse the differences and possible improvements over this work.  How does the time dependent infection rate approach obtained through PINNs in this earlier work compares with the procedure used by the authors? 

Reviewer 2 Report

In this paper, the authors propose a neural network-based approach to estimate COVID-19 epidemiologic parameters also addressing the issue of identifiability. They also propose a method of de-noising epidemiologic data. I have some concerns regarding the paper in its present form:
Major:
1.    The paper is very technical without having a clear real world application since the underlying model is too simple and only of theoretical interest. It ignores major important aspect of the epidemic such as vaccination, age-stratification, virus variants or immune waning. Maybe a machine-learning journal is better suited to publish this research.
2.    I miss a major topic or central theme in the paper. It is a bit of addressing noise in data, identifiability of parameters in ODEs, parameter estimation with neural networks and interpretation of data and parameter estimates. I would suggest focussing on one of the issues in more detail and to provide a clear conclusion or recommendation.
3.    The authors emphasize the importance of death data for parameter estimation. This might be true from a theoretical point of view, but I would challenge this assumption in a real world situation since COVID-19 death data are highly biased. A few issues are (1) clear underreporting in view of general death statistics, (2) large reporting delays, (3) unknown reasons for deaths (i.e. it is very difficult to distinguish “died with COVID-19” and “died because of COVID-19” since autopsy data are rarely available). The same applies for the number of hospitalizations, which, for example, depend on vaccination status, virus variants or advances in COVID-19 treatment.
4.    The authors aimed at denoising the data with a wavelet approach. However, I would challenge the assumption of random fluctuations of the data. Rather than this, the data are subjected to strong systematic biases, which could be time dependent too (see #3). This cannot be corrected with the chosen approach.
5.    Line 378: The authors mention issues with the scaling of the parameters. This is strange since it is recommended to rescale all parameters prior to network learning attempts. Overall, the learning approach is not provided into sufficient detail, e.g. with respect to network architecture or pruning efforts.
6.    The authors demonstrated that their data could be learned but this is not surprising in the context of neural network learning. I miss the issue of validation of the model and considerations of over-fitting of the neural network.
7.    Lines 536-559: These interpretations are highly speculative and could be modelling and data artefacts. It is known for example that the dark figure depends on the dynamics of the epidemic itself, i.e. in case of high infection numbers, the under-reporting typically increases.
8.    Overall, the paper is not very well organized. There are discussions in the methods. There are methods and discussions in the results.

Minor:
1.    Major figures are not very informative, e.g. it is not clear what could be learned from figure 3.
2.    Page 13/14: Parameter values should be provided with a reasonable number of digits.
3.    The paper requires language polishing.

Author Response

Please see attached letter

Reviewer 3 Report

The authors of the paper describe their proposed approach for A Modified PINN Approach for Identifiable Compartmental Models in Epidemiology with Application to COVID-19. The topic is interesting and with possible applicability. However, the paper needs several improvements:

1) the main contribution and originality should be explained in more detail

2) the motivation of the approach with NNs needs further clarification, why not other methods?

3) discussion of related work in COVID-19 should be expanded with more recent work

4) Minor grammar and syntax issues need correction

5) more simulation results and formal comparison of results are needed

6) the conclusions should be extended with more future work

7) The names of tables should be on top, not below

8) The numbering of equations is of the form 1, 2, ...,n, not by section (2.1, 2.2, etc.)

9) More references to COVID-19 papers should be included, like:

Prediction of COVID-19 active cases using exponential and non-linear growth models. Expert Syst. J. Knowl. Eng. 39(3) (2022)

Design of Type-3 Fuzzy Systems and Ensemble Neural Networks for COVID-19 Time Series Prediction Using a Firefly Algorithm. Axioms 11(8)410 (2022)

Interval type-3 fuzzy aggregators for ensembles of neural networks in COVID-19 time series prediction. Eng. Appl. Artif. Intell. 114105110 (2022)

Round 2

Reviewer 1 Report

The authors modified their text and made improvements.  I believe the paper can be published.  I have the following comments:

1. I do not agree with the authors regarding the restrictive nature of the linear  a(t) assumption in the new ref 29.  If you look at Fig. 7 of the last version of the present paper you can clearly see an average linear drop followed by an average linear increase.

2. The Figure 1 of the present work is conspicuously similar to the Figure 1 of ref. 29.  I think it would be best for the present work to alter the figure somehow.

3. The added reference 29, of which the present work is  essentially a follow up and extension with a more complex model, is not even correctly stated in the reference section.  The authors should be more careful with this issue.

Author Response

Dear Reviewer,

We have attached the reply letter here, as well as the revised paper. 

Thanks for your time and effort in reviewing our paper.

Hongkun Zhang (on behalf of all the authors for this paper)

Reviewer 3 Report

The authors have made all the suggested modifications and the paper can be accepted.

Author Response

Thanks for your recommendations. As there have been no requirement to make any changes from the third reviwer. We do not attach a file here.